# Feeling Connected: Technology-Mediated Communication and the Relationship between Modality and Affective Outcomes

**Tamara J. Skootsky * , Diana R. Sanchez and Kentaro Kawasaki**

Psychology Department, San Francisco State University, 1600 Holloway Avenue, Ethnic Studies & Psychology Building, San Francisco, CA 94132, USA; sanchezdianar@sfsu.edu (D.R.S.)

\* Correspondence: tskootsky@gmail.com

**Abstract:** The relationship between workplace communication and affective outcomes, specifically connectedness at work and affective organizational commitment, is one that warrants further investigation for practical usage in the increasingly multimodal workplace. This study considers the frequency of use across five communication modalities, that being face-to-face, email, phone calls, instant messaging, and video calls, in relation to affective outcomes, as well as their relationships with communication meaningfulness. Employed participants ($n$ = 516) completed an online survey in which they self-reported weekly communication tendencies, experienced connectedness, and affective organizational commitment. The final sample consisted of participants across 20 different industries in the United States. The most respondents worked in Health Care and Social Assistance or Professional, Scientific, and Technical Services (15% each), followed closely by respondents from Educational Services or Finance and Insurance (13% each). Data collection took place in between October 2021 and January 2022. Generally, participants who reported more frequent communication at work reported higher levels of connectedness and higher affective organizational commitment. Employees who found their communication more meaningful (irrespective of frequency) felt the most emotionally connected. Unique benefits of different communication modalities, as well as implications for hybrid and remote organizations, are discussed.

**Keywords:** technology-mediated communication; workplace technology; connectedness at work; loneliness; affective organizational commitment; meaningfulness; media richness

## 1. Introduction

Technology-mediated communication has become the norm in the workplace, often using high bandwidth video conferencing and dynamic chat software [1,2] in place of face-to-face conversations. Over 60% of employees report working on at least one virtual team [3,4] and even when co-located (in the same building as their coworkers), they use communication mediated by technology [5].

Researchers have noted that non-verbal signals may be lost in technology-mediated conversations (e.g., facial or body expressions, physical proximity cues), while new technologically unique signals are generated (e.g., time-lags, typos) [6–8].

Communication modality can impact emotional reactions (i.e., affective outcomes) that employees have to technology-based communication when compared to both in-person interactions and compared with different types of technology-mediated communication [9–11]. Understanding how different types of technology-based communication impact affective outcomes at work, could help employees and organizations manage communication effectively, build stronger coworker relationships and potentially improve perceptions of job significance [12], learning outcomes [13], and buffer against racial discrimination [14].

This study aims to understand how different technological modalities used for communication at work, impact affective employee experiences at the individual level (i.e.,

workplace connectedness [15]) and organizational level (affective organizational commitment [16]). Through correlational research, this study aims to contribute novel understanding to whether and how choices about communication technology modality impact employees at the affective and interpersonal level. Because these daily choices regarding which technologies to use may have downstream effects, it is important for managers, employees and organizations to be equipped with the information necessary to make choices that are best suited to their needs.

### 1.1. Communication Modality Impacts Communication Outcomes

While work-related outcomes, such as productive output [10], communication quality [17], task focus [18] and efficiency [19], improve with the use of technology-mediated communication tools, the use of these tools may also reduce social anxiety [20], increase voice among minority group members [21,22] and decrease the salience of visual biases [23]. Despite the variety of these benefits for using technology-mediated communication tools, the loss of socially relevant non-verbal cues can detract from social bonding and interpersonal connections [6,24]. While individuals may improve their relational skills with greater use of any communication medium, the overall constraints of workplace priorities and culture may not be conducive to the structures and time necessary to develop or select stronger virtual connections [25–27].

### 1.2. Communication Modality and Connectedness

Connectedness is a subjective emotional state characterized by a supply of desired social connections, including those that provide adequate emotional support, shared activities, and interests [15]. In contrast, experienced loneliness would mean there is a discrepancy between desired and actual feelings of social connectedness [28]. Individuals who feel lonely may report depression and social anxiety [29], as well as severe impacts on health [30] which can negatively affect decision-making and lead to withdrawal behaviors [15].

Technology-mediated communications may increase connectedness as people are more in touch with others compared to individuals who have no interpersonal communication [31,32]. However, face-to-face interactions tend to generate the most positive interpersonal outcomes [33,34]. Much remains to be explored about the impact of different technology-mediated communication tools on experienced connectedness and loneliness. Further understanding of this may lead organizations, managers, and their employees to develop strategies for choosing and using technological tools over time to maintain communication.

### 1.3. Communication Modality on Affective Organizational Commitment

Organizational commitment represents the extent to which a person identifies with, feels connected to, or feels uncomfortable leaving an organization [35]. This affective commitment is conceptualized as a psychological bond or affiliation with an organization that arises because of effective socialization within the organization [36,37]. Organizational commitment has been a long-studied construct in organizational psychology because of its relationship to worker satisfaction, effort expenditure, and turnover, among numerous other organizationally relevant outcomes [37]. When employees lack interpersonal attachments within an organization, they may report decreases in organizational commitment [38].

Organizational commitment is positively related to communication in any form, which may be a result of increased communication satisfaction [39,40]. Research suggests that greater use of employee portals [41] and email communications [42] can positively predict organizational commitment, although modality itself may explain a small percentage of the variance found [41].

Research has demonstrated that organizational communication impacts employee organizational commitment [42]. Furthermore, evidence supports the idea that certain technology-mediated communication modalities may influence this relationship [42]. As organizational commitment is a known predictor of turnover intentions, job satisfaction, and

effort expended towards work [37], it is important to understand whether this organization-level construct may be influenced by the rise of technology-driven communication practices in the modern workplace.

### 1.4. Media Characteristics Perspective

The media characteristics or cues-filtered-out [43] perspective of technology-mediated communication considers communication modalities based on the signaling channels or affordances they offer. These theories contend that fewer non-verbal channels result in less connective interpersonal interactions [43]. Two prominent theories within this literature are social presence theory [7] and media richness theory [44].

Social presence theory [7] suggests that modalities vary in their ability to generate the sense of another's presence. Social presence is a subjective judgement, which varies at the individual level. Early studies of perceived social presence found consistent rankings of communication modalities across individuals such that face-to-face was perceived as having the most social presence, and written communication having the least [7].

Media richness theory [44] proposes that media differ in their efficacy in transmitting certain types of information but additionally considers whether communication modality is adequately matched with communication goals. This theory considers the extent to which information being communicated is known versus unknown (uncertainty) and whether there is one or multiple possible answers (equivocality). It ranks communication media concretely by their respective capacities for feedback, cues, personalization, and language variety. Because equivocality is endemic to interpersonal communication, it is expected that richer modalities would be better suited to building and maintaining relationships.

Both theories are founded on the idea that more communication affordances, benefit interpersonal, uncertain, or complex message transmission while fewer cues are better optimized for relaying concrete, standardized or simple task-related information [7,44].

Media use and outcomes in natural settings are likely also derived from social contexts, and thus, to understand the effects of media modality in organizations, we go beyond media characteristics and include communication content and personal meaningfulness by modality.

We will evaluate both work-related communication as well as non-work communication. We will examine non-work-related communication because of its demonstrated relationship with affective organizational commitment [45], and meaningfulness because of its subjectivity and demonstrated relationship with communication modality in long-distance relationships [46].

### 1.5. Hypotheses

The effects of greater communication in any modality, especially when given sufficient time to develop trust and communication norms, generally result in positive interpersonal outcomes [27,33,47]. Therefore, we hypothesize that frequent use of any communication modality will result in more positive affective outcomes for employees, such as greater connectedness and increased organizational commitment. Specific to this study, communication modalities of interest will include face-to-face communication, email, phone calls, instant messaging, and video calls.

While these outcomes may be found across communication modalities, the content and interpretation of communication processes may influence the feelings of connection that an individual has with their conversation partners, and the organization, as a result.

**H1.** *Communication frequency (regardless of modality type) will be significantly related to workplace connectedness (H1a) and affective organizational commitment (H1b).*

**H2.** *Technological modalities rated as more meaningful will be significantly related to workplace connectedness (H2a) and affective organizational commitment (H2b).*

We further seek to understand the relationship between modality use and the contextual factors of this study.

- Is the frequency of work and non-work-related communication related?
- Is the type of modality common between work and non-work-related communication tendencies?
- Is the frequency of a modality used for communication related to the perceived meaningfulness of that modality?
- Are there latent profiles that explain modality use of individuals?

The timing of the research is focused on the behaviors of individuals in a post pandemic workplace, and data collection will take place between October 2021 and January 2022.

## 2. Materials and Methods

### 2.1. Power Analysis

Prior studies of communication modality on outcomes of connectedness/loneliness were used to estimate the required sample size [33].

We expect correlations between communication frequency and connectedness to between r = 0.10 and 0.26 [33]. A power analysis conducted using G*Power version 3.1.9.7 with an alpha level of 0.05 suggested that a minimum of 186 participants would be needed to detect this effect at the highest range, and 1293 participants necessary to detect the smaller effects. The difference between modalities was expected to be between r = 0.20 and 0.35, requiring 178 to 542 participants to detect the effects.

### 2.2. Participants

Based on the power analysis, five hundred and forty-two participants were required. All participants were recruited from Prolific Academic, an online tool for recruiting research participants. Prolific Academic was chosen as it is a large online crowdsourcing community for recruiting participants for various types of scientific studies. All participants that met the eligibility criteria, that being working 30+ h a week for one employer and being at least 18 years old, were included in the data sample. To ensure critical demographic information about various working situations are captured, participants would indicate their job title, estimate how many hours per week on average they spend at work, the frequency with which they worked remotely, and the average number of hours per week the participant spends in communication with others. After confirming eligibility, participants were directed to a survey completion tool to complete the survey.

### 2.3. Procedure

Each participant accessed the survey online via Prolific Academic. After viewing a consent form and confirming they met the study requirements, each participant completed the survey which was comprised of the measures listed below. Scale items were randomly presented, and the order of the scales was counterbalanced across participants. Throughout the survey, attention check questions were included to help identify and dissuade insufficient effort responding [48]. At the end of the survey, participants viewed a debrief statement. Based on pilot testing, we identified the survey would take approximately 6 min to complete. Participants were paid $0.70 for their time.

### 2.4. Measures

Prior research and results from a pilot test were reviewed to select validated tools for measuring the constructs of interest for this study. Validated tools relevant to modality use, types of communication, meaningfulness, connectedness, and affective organizational commitment were researched and adapted for the purposes of the study. These scales were then collected and used to create a survey for participants to answer. Participants were asked to consider their current job when responding to the survey. The survey used for this study is provided as part of the Supplementary Materials.

### 2.4.1. Demographic and Control Measures

Participants reported demographic information including age, gender, ethnicity, work experience with current organization, industry, total number of hours worked per week, and total remote hours worked per week.

### 2.4.2. Frequency of Modality Use

Communication modality refers to the type or channel of communication, especially in virtual environments [49]. In this study, the modalities of face-to-face, email, phone calls, instant messaging (IM), and video calls were studied. In previous related studies, five-point Likert type scales have been used to measure communication frequency [33,46], which was the method we applied in the current study. Participants were asked to estimate the frequency of their communication for both work-related communication and non-work-related communication.

Non-work communication, or informal discussion at work, includes all interactions about personal topics used to build and maintain social relationships and connections that are not directly related to work activities [50]. Non-work communications were drawn from social media research studies [51]. To measure non-work communication in this study we drew from Oksa, Kaakinen and colleagues' 2020 work on enterprise social media use [51], using a 5-point Likert-type scale ranging from "I don't use it" to "Many times a day".

### 2.4.3. Meaningfulness

For meaningfulness of communication by modality, we wrote a single-item 5-point Likert type question modeled after familiarity and awareness scales [46]. Consulting existing studies on the subject with a focus on meaningfulness and its related subjectivity, as well as existing studies with similarly subjective words with similar meanings such as "important", "worthwhile", and "valuable", our single-item question was constructed with careful inclusion of introductory verbiage which emphasized the subjectivity of meaningfulness [46] and included both "valuable" and "important" in the definition [52,53].

### 2.4.4. Connectedness

Loneliness is a complex emotional state that arises from feeling distant from or unable to connect with others [15]. The 16-item loneliness at work scale was used to measure workplace loneliness [54]. The scale has two subfactors, wherein only one of the subscales is included in this analysis. The 7-item subfactor, social companionship, was the most relevant to the purposes of this study, and was measured on a 7-point Likert type scale ranging from Strongly Disagree to Strongly Agree. In our data, the Cronbach's alpha for this measure was $\alpha = 0.90$, respectively. An example item is, "I often feel isolated when I am with my coworkers". A higher score represents a stronger disconnection with coworkers.

### 2.4.5. Affective Organizational Commitment

Organizational commitment is a psychological construct representing how much an employee feels dedicated to their place of work [16]. In this study, organizational commitment was measured using the 8-item Affective Organizational Commitment Scale [16]. It is measured on a 7-point Likert type scale ranging from Strongly Disagree to Strongly Agree. In our sample, the reliability for this scale was $\alpha = 0.86$. An example item is, "This organization has a great deal of personal meaning for me". A higher score represents a stronger feeling of affective organizational commitment.

## 3. Results

### 3.1. Pilot Tests

Prior to data collection, two pilot tests (n = 26, then n = 23) were used to review the survey for errors and to estimate the duration of the survey.

### 3.2. Tools for Analysis

Analyses were completed using the statistical program *R* version 4.3.2 [55] with the stats package and functions cor, manova, and aov. We also used mclust via the tidyLPA wrapper, to estimate and select the latent profiles [56].

### 3.3. Sample Demographics

Five hundred and twenty-five participants completed this study. We removed three responses that failed more than 2/3 of the in-survey attention checks [48], and an additional four responses which contained missing or abnormal data entries. The final sample consisted of five hundred and sixteen total participants across 20 different industries, see Table 1.

**Table 1.** Breakdown of Total Sample (n = 516) by Industry Representation.

| Industry | n | Percent |
|---|---|---|
| Professional, Scientific, and Technical Services | 76 | 14.7% |
| Health Care and Social Assistance | 75 | 14.5% |
| Educational Services | 63 | 12.2% |
| Finance and Insurance | 60 | 11.6% |
| Government | 35 | 6.8% |
| Information | 35 | 6.8% |
| Arts, Entertainment, and Recreation | 34 | 6.6% |
| Other Services (except public administration) | 28 | 5.4% |
| Retail Trade | 21 | 4.1% |
| Administrative and Support Services | 20 | 3.9% |
| Manufacturing | 16 | 3.1% |
| Real Estate and Rental and Leasing | 14 | 2.7% |
| Accommodation and Food Services | 7 | 1.4% |
| Agriculture, Forestry, Fishing, and Hunting | 6 | 1.2% |
| Construction | 6 | 1.2% |
| Management of Companies and Enterprises | 6 | 1.2% |
| Transportation and Warehousing | 5 | 1.0% |
| Utilities | 5 | 1.0% |
| Wholesale Trade | 3 | 0.6% |
| Mining, Quarrying, and Oil and Gas Extraction | 1 | 0.2% |

Survey respondents averaged M = 32 years of age (SD = 10 years), had been employed at their current organization for an average M = 5 years (SD = 5) and worked an average M = 39.2 h a week (SD = 8.57). Most respondents were White (73%) followed by Asian (8%), mixed-ethnicity (8%), Black (6%) and Hispanic/Latinx (4%). Participants of all gender orientations were represented, with the majority identifying as either Male (22%) or Female (76%).

### 3.4. Hypothesis Testing

Responses for many of the variables were correlated within the expected range 0.10–0.20 or higher, see Table 2. A Bonferroni adjustment was applied to the correlation table to ensure conservative results. Because non-work communication and meaningfulness ratings were only collected for modalities used by a given participant, pairwise correlations are reported in the table, meaning that sample sizes will vary accordingly.

**Table 2.** Descriptive Statistics and Correlations for Study Variables.

| Variable | M | SD | 1 | Individual Characteristics 2 | 3 | 4 | Study Outcomes 6 | 7 | 8 | 9 | Work-Related Communication 10 | 11 | 12 | 13 | 14 | 15 | Non-Work-Related Communication 16 | 17 | 18 | 19 | 20 | 21 | Meaningfulness Rating 22 | 23 | 24 |
|---|---|---|---|---|---|---|---|---|---|---|---|---|---|---|---|---|---|---|---|---|---|---|---|---|---|
| **Individual Characteristics [4]** | | | | | | | | | | | | | | | | | | | | | | | | | |
| 1. Age | 32.3 | 9.52 | | | | | | | | | | | | | | | | | | | | | | | |
| 2. Weekly Hours Worked | 39.1 | 8.57 | 0.13 | | | | | | | | | | | | | | | | | | | | | | | |
| 3. Remote Work [1] | 1.47 | 0.50 | −0.20 | −0.10 | | | | | | | | | | | | | | | | | | | | | | |
| 4. Work Experience | 4.53 | 5.20 | 0.53*** | 0.16 | 0.03 | | | | | | | | | | | | | | | | | | | | | |
| **Study Outcomes [2,4]** | | | | | | | | | | | | | | | | | | | | | | | | | |
| 6. Connectedness | 3.15 | 1.33 | <0.01 | −0.13*** | 0.23*** | −0.10 | | | | | | | | | | | | | | | | | | | | |
| 7. Aff. Org. Commitment | 4.08 | 1.37 | 0.17 | 0.04 | −0.10 | 0.17* | −0.49*** | | | | | | | | | | | | | | | | | | | |
| **Work-Related Communication [3,4]** | | | | | | | | | | | | | | | | | | | | | | | | | |
| 8. Face-to-face | 4.45 | 8.24 | −0.10 | 0.06 | −0.49*** | 0.01 | −0.21*** | 0.13 | | | | | | | | | | | | | | | | | | |
| 9. Email | 5.82 | 7.09 | <0.01 | 0.23*** | <0.01 | 0.01 | −0.10 | <0.01 | 0.01 | | | | | | | | | | | | | | | | | |
| 10. Phone calls | 3.73 | 6.56 | <0.01 | 0.19* | 0.09 | −0.10 | <0.01 | −0.10 | <0.01 | 0.19** | | | | | | | | | | | | | | | | |
| 11. IM | 4.86 | 7.40 | −0.20 | 0.16 | 0.10 | −0.10 | −0.10 | 0.03 | 0.07 | 0.35*** | 0.20** | | | | | | | | | | | | | | | |
| 12. Video calls | 3.65 | 5.22 | 0.01 | 0.24*** | <0.01 | <0.01 | −0.10 | −0.10 | −0.10 | 0.18* | <0.01 | 0.08 | | | | | | | | | | | | | | |
| 13. Total | 17.6 | 12.0 | −0.10 | 0.39*** | −0.20 | <0.01 | −0.22*** | <0.01 | 0.39*** | 0.44*** | 0.45*** | 0.42*** | 0.36*** | | | | | | | | | | | | | |
| **Non-Work-Related Communication [3]** | | | | | | | | | | | | | | | | | | | | | | | | | |
| 14. Face-to-face [5] | 2.48 | 1.24 | 0.02 | −0.10 | <0.01 | 0.07 | −0.20 | 0.09 | 0.17 | 0.02 | 0.08 | 0.14 | <0.01 | 0.10 | | | | | | | | | | | | |
| 15. Email [6] | 1.47 | 1.39 | 0.13 | 0.02 | −0.10 | 0.17 | 0.01 | 0.06 | 0.09 | 0.02 | 0.08 | <0.01 | 0.12 | 0.08 | 0.35*** | | | | | | | | | | | |
| 16. Phone calls [7] | 1.59 | 1.26 | 0.07 | 0.03 | <0.01 | 0.08 | −0.10 | 0.07 | 0.13 | 0.06 | 0.09 | <0.01 | 0.06 | 0.13 | 0.38*** | 0.59*** | | | | | | | | | | |
| 17. IM [8] | 2.48 | 1.41 | <0.01 | 0.04 | −0.10 | <0.01 | −0.23** | 0.08 | 0.07 | 0.01 | 0.02 | 0.11 | 0.09 | 0.11 | 0.39** | 0.40*** | 0.47*** | | | | | | | | | |
| 18. Video calls [9] | 1.07 | 1.06 | <0.01 | 0.12 | −0.10 | <0.01 | −0.10 | 0.05 | 0.07 | 0.02 | 0.05 | 0.07 | 0.28*** | 0.13 | 0.16 | 0.50*** | 0.46*** | 0.33*** | | | | | | | | |
| 19. Average [10] | 1.25 | 0.86 | <0.01 | 0.16 | −0.35*** | 0.10 | −0.28*** | 0.14 | 0.29*** | 0.13 | 0.13 | 0.11 | 0.19* | 0.28*** | 0.60*** | 0.72*** | 0.74*** | 0.70*** | 0.64*** | | | | | | | |
| **Meaningfulness Rating [2]** | | | | | | | | | | | | | | | | | | | | | | | | | |
| 20. Face-to-face [5] | 4.02 | 1.10 | <0.01 | <0.01 | 0.05 | 0.03 | −0.20 | 0.17 | 0.15 | 0.06 | 0.11 | 0.07 | 0.16 | 0.18 | 0.49*** | 0.25 | 0.31* | 0.31* | 0.21 | 0.44*** | | | | | | |
| 21. Email [6] | 2.31 | 1.23 | 0.24*** | 0.01 | <0.01 | 0.19* | −0.10 | 0.19* | 0.07 | 0.03 | <0.01 | <0.01 | <0.01 | 0.02 | 0.24 | 0.54*** | 0.28*** | 0.23*** | 0.32*** | 0.39*** | 0.23 | | | | | |
| 22. Phone calls [7] | 2.94 | 1.33 | 0.09 | 0.06 | <0.01 | 0.12 | −0.10 | 0.18 | 0.11 | 0.04 | 0.03 | 0.01 | 0.06 | 0.11 | 0.24 | 0.35*** | 0.51*** | 0.36*** | 0.31*** | 0.46*** | 0.52*** | 0.49*** | | | | |
| 23. IM [8] | 3.18 | 1.25 | 0.10 | <0.01 | 0.04 | 0.02 | −0.27*** | 0.15 | 0.08 | <0.01 | −0.10 | 0.10 | 0.08 | 0.09 | 0.33* | 0.25** | 0.26** | 0.61*** | 0.19 | 0.40*** | 0.48*** | 0.41*** | 0.53*** | | | |
| 24. Video calls [9] | 2.81 | 1.34 | <0.01 | 0.01 | <0.01 | −0.10 | −0.10 | 0.16 | 0.10 | <0.01 | <0.01 | 0.01 | 0.23** | 0.11 | 0.16 | 0.30*** | 0.33*** | 0.36*** | 0.56*** | 0.43*** | 0.45*** | 0.40*** | 0.62*** | 0.40*** | | |
| 25. Average [10] | 1.69 | 0.85 | 0.02 | 0.19*** | −0.43*** | 0.15 | −0.39*** | 0.26*** | 0.39*** | 0.12 | 0.12 | 0.08 | 0.07 | 0.31*** | 0.34*** | 0.40*** | 0.41*** | 0.45*** | 0.32*** | 0.74*** | 0.64*** | 0.55*** | 0.71*** | 0.58*** | 0.44*** | |

[1] 0 = Partially remote, 1 = Fully remote. [2] Reported on a 5-point scale. [3] Reported in hours. [4] n = 516. [5] n = 245. [6] n = 471. [7] n = 340. [8] n = 404. [9] n = 387. [10] n = 516. * $p < 0.05$, ** $p < 0.01$, *** $p < 0.001$.

Hypothesis 1 was partially supported. For H1a total work-related communication and average non-work-related communication were both significantly related to connectedness, but H1b was not supported as neither total work communication nor average non-work communication were related to affective organizational commitment. When focusing on work-related communications, it appears that more communication is related to a sense of connectedness but not affective organizational commitment. This holds true for non-work-related communication as well.

Hypothesis 2 was fully supported as average meaningfulness ratings were significantly related to both connectedness (supporting H2a) and affective organizational commitment (supporting H2b). We can infer from these results that people who find their workplace communication overall to be meaningful feel more connected to their organization and the people in it.

### 3.5. Exploratory Analyses

In reviewing results for specific modalities to explore our questions, we found that for work-related communication, only the frequency of face-to-face communication was significantly related to connectedness. For non-work-related communication, only IM was significantly related to connectedness. For meaningfulness, more meaningful IMs were related to connectedness whereas more meaningful emails were related to affective organizational commitment.

Non-work communication over video, that being informal or work unrelated communication, is influenced by the total hours reported on video calls ($r = 0.28$, $p < 0.001$, $n = 367$) and is the only distinct modality to have a linear effect with its own outcomes. Within the sample of five hundred and sixteen participants, the data showed that participants worked an average $M = 39.2$ h a week ($SD = 8.57$), with an average of $M = 17.6$ h being work-related communication ($SD = 12.00$) whereas $M = 1.25$ h was non-work-related communication ($SD = 0.816$). The average of non-work communication frequency across modalities ($n = 516$ for all) is influenced by raw frequency of face-to-face communication ($r = 0.29$, $p < 0.001$), video calls ($r = 0.19$, $p = 0.015$) and total hours spent communicating each week ($r = 0.28$, $p < 0.001$). Uniquely from other forms of communication, an increased number of video calls increases the chance that they will include some form of informal conversation in that modality but may not "spill over" into other forms. The overall results align with our expectation that when there is more time to communicate, the possibility for non-work versus task-oriented-only communication increases. In addition, workplaces characterized by, or individuals who choose, in-person work (face-to-face) and/or video calls may have work that is structurally or culturally more conducive to non-work-related conversation than others.

Like non-work-related communication, increased use of video calls correlates significantly with higher reports of meaningfulness in that modality only ($r = 0.23$, $p = 0.007$, $n = 367$). Additionally, the average meaningfulness rating across all modalities ($n = 516$) for respondents is significantly influenced by both face-to-face ($r = 0.39$, $p < 0.001$) and overall communication frequency ($r = 0.31$, $p < 0.001$). While frequent video callers may become more adept at finding ways to connect meaningfully in that modality specifically, individuals who connect more with their coworkers in-person, or in general, may find that they have more meaningful interactions overall.

### 3.6. Latent Profiles of Modality

In a latent profile analysis, a four-profile solution (see Figure 1) was selected based on an Analytic Hierarchy Process which considers five information criteria [56–58]. This study reports the local solution for our data set [58].

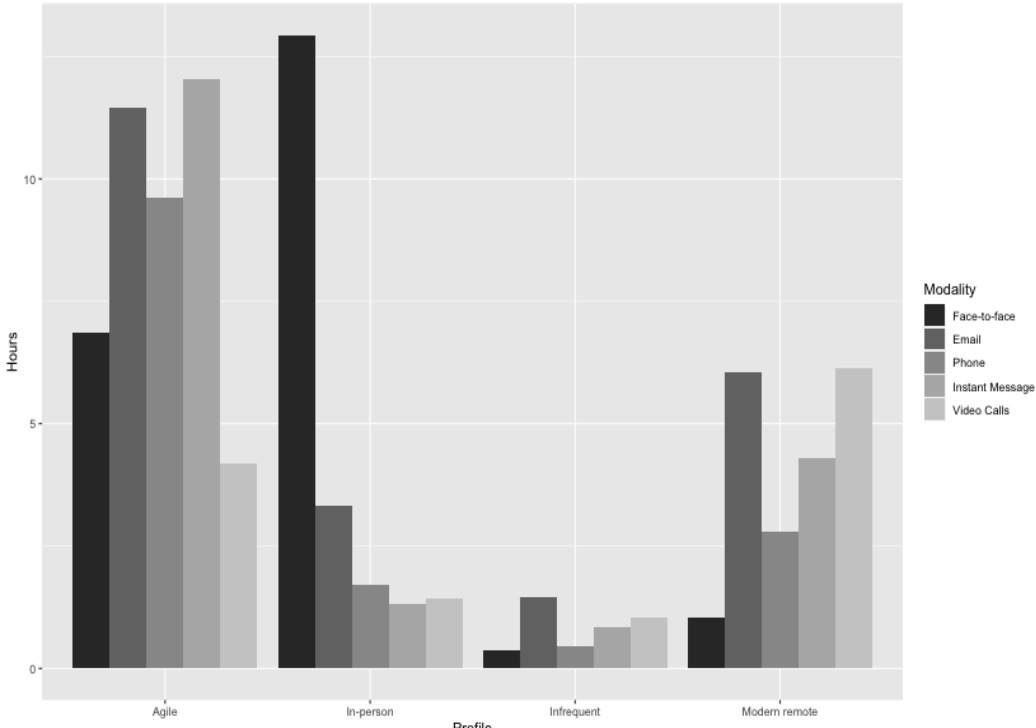

**Figure 1.** Modality Use of the Four Communicator Profiles Extracted.

The first extracted group was large (n = 179, 34.6%) and showed low use of face-to-face communication, high use of video calls, and moderate use of other virtual technologies. We characterize these people as Modern Remote Communicators The second extracted group was moderately large (n = 126, 24.4%) and showed moderate to high usage of each of the five modalities studied. These people can be characterized as Agile Communicators, demonstrating an ability to switch between modalities. The third extracted group was also moderately large (n = 118, 22.9%) and showed low to no use of any communication modalities. These people can be characterized as Infrequent Communicators. Finally, the fourth extracted group was moderately small (n = 93, 18%) and can be characterized as In-person Communicators as they have the highest frequency of face-to-face communication, but low frequency of use for all virtual modalities. Once extracted, these four profiles were merged with the original data set and used as a categorical factor to test outcomes across groups. Below, these results are discussed of outcomes by modality, although a more rigorous approach could be to consider these as criteria validation of the profiles themselves.

A multivariate analysis of variance (MANOVA) was conducted to determine whether the modality use profiles could differentiate outcomes related to connectedness at work and affective organizational commitment. Modality use profile significantly predicted the combination of affective outcomes in this study, $F_{approx}(3, 512) = 6.01$, $p < 0.001$, Pillai's V = 0.10, $\eta^2 = 0.03$.

The first stepdown analysis, conducted using analysis of variance (ANOVA), showed a significant effect of modality use profile on connectedness, $F(3, 512) = 16.22$, $p < 0.001$, $\eta^2 p = 0.09$. A Bonferroni-adjusted post hoc test showed that, specifically, Infrequent Communicators ($M_{adj} = 3.83$, SE = 0.12) reported higher scores of disconnection as compared to Modern Remote Communicators ($M_{adj} = 3.09$, SE = 0.10), $p < 0.001$, d-hat = −0.58, Agile Communicators ($M_{adj} = 2.92$, SE = 0.11), $p < 0.001$, d-hat = −0.72, and In-Person Communicators ($M_{adj} = 2.73$, SE = 0.13), $p < 0.001$, d-hat = 0.87. This provides further support for our initial hypothesis that any pattern of communication is important and valuable for feeling like one is part of a social network. Given the nature of LPA, this could also suggest that the first three profiles do not significantly distinguish from each other theoretically.

An analysis of covariance (ANCOVA) demonstrated that modality use profiles also account for significant variance in affective organizational commitment when controlling for workplace connectedness, F(3, 510) = 5.66, $p < 0.001$, $\eta^2 p = 0.002$. However, a Bonferroni-adjusted post hoc test found no significant pairwise comparisons.

A second multivariate analysis of variance (MANOVA) was conducted to determine whether modality use profile was related to frequency of non-work communication or meaningfulness. Modality use profile significantly predicted the combination of alternate predictor variables in this study, $F_{approx}(3, 512) = 15.90$, $p < 0.001$, Pillai's V = 0.17, $\eta^2 = 0.09$. The first stepdown analysis, conducted using analysis of variance (ANOVA), showed a significant effect of modality use profile on non-work communication, F(3, 512) = 15.33 $p < 0.001$, $\eta^2 p = 0.08$. A Bonferroni-adjusted post hoc test showed that, specifically, Infrequent Communicators ($M_{adj} = 0.82$, SE = 0.08) reported lower scores of non-work communication frequency as compared to Modern Remote Communicators ($M_{adj} = 1.27$, SE = 0.06), $p < 0.001$, d-hat = 0.54, Agile Communicators ($M_{adj} = 1.47$, SE = 0.07), $p < 0.001$, d-hat = 0.77, and In-Person Communicators ($M_{adj} = 1.47$, SE = 0.09), $p < 0.001$, d-hat = −0.78.

The second stepdown analysis, conducted using analysis of covariance (ANCOVA), showed that modality use profile accounted for a significant portion of the variability in subjective meaningfulness, after controlling for non-work communication frequency, F(3, 511) = 66.53, $p < 0.001$, $\eta^2 p = 0.09$. A Bonferroni-adjusted post hoc test showed significant differences between every communication profile except *Modern Remote Communicators* ($M_{adj} = 1.65$, SE = 0.04), and Agile Communicators ($M_{adj} = 1.78$, SE = 0.05), which both report more meaningfulness, on average, than Infrequent Communicators ($M_{adj} = 1.43$, SE = 0.05), and lower meaningfulness, on average, than In-Person Communicators ($M_{adj} = 1.98$, SE = 0.06, see Table 3 for pairwise contrasts).

**Table 3.** ANCOVA post hoc estimates for pairwise contrasts of modality use profiles on subjective meaningfulness, holding non-work communication frequency constant.

| Contrast | Mean Differences | | | | | Effect Sizes | | | |
|---|---|---|---|---|---|---|---|---|---|
| Profiles [1] | df | Est. | SE | T-Ratio | *p*-Value | Est. | SE | Lower Cl | Upper Cl |
| 1-2 | 511 | −0.13 | 0.06 | −2.02 | 0.2674 | −0.24 | 0.12 | −0.47 | −0.01 |
| 1-3 | 511 | 0.22 | 0.07 | 3.28 | 0.0067 | 0.40 | 0.12 | 0.16 | 0.64 |
| 1-4 | 511 | −0.33 | 0.07 | −4.76 | <0.0001 | −0.61 | 0.13 | −0.87 | −0.36 |
| 2-3 | 511 | 0.35 | 0.07 | 4.76 | <0.0001 | 0.63 | 0.13 | 0.37 | 0.90 |
| 2-4 | 511 | −0.21 | 0.07 | −2.74 | 0.038 | −0.38 | 0.14 | −0.64 | −0.11 |
| 3-4 | 511 | −0.55 | 0.08 | −7.05 | <0.0001 | −1.01 | 0.15 | −1.29 | −0.72 |

[1] 1 = Modern Remote Communicators, 2 = Agile Communicators, 3 = Infrequent Communicators, 4 = In-Person Communicators. *p*-values adjusted with Bonferroni method for six tests.

## 4. Discussion

The obtained patterns of data suggest that connection was influenced by the most factors, including face-to-face frequency, overall frequency, non-work communication frequency, average subjective meaningfulness, and meaningfulness of instant message communication. Average meaningfulness and email meaningfulness related to affective organizational commitment.

Four profiles of communication styles were identified, with some distinct outcomes for certain profiles. Infrequent Communicators had the poorest interpersonal outcomes, with significantly higher disconnection, less frequent non-work communication, and the lowest subjective meaningfulness ratings of those communications. Conversely, In-Person Communicators had the highest meaningfulness ratings overall, while virtual communicators, both Agile Communicators and Modern Remote Communicators, had intermediate meaningfulness ratings statistically distinct from both the highest and lowest groups but not different from each other.

### 4.1. Effects of Face-to-Face Communication

This result largely aligns with our initial literature review, in which multiple studies distinguished between functional facets of isolation in comparison to psychological isolation [33,38]. Prior loneliness research suggested that greater frequency of any of our predictors would be better for these outcomes [31,32], which they were in our study as well.

In addition, prior research showed that face-to-face communication tends to have unique benefits for affective outcomes [6,24]. Both average non-work communication and average meaningfulness were affected by average total frequency of communication and average frequency of face-to-face communications, which hints at a pattern in our data where more communication is better and face-to-face specifically is better for connection. Frequency of video calls is also related to average non-work communication in our results, suggesting that modern video calling may be able to compensate for some of what is lost from technology-mediated connection.

### 4.2. Benefits of Instant Messaging

Social support literature may explain the unique relationship between instant message communication and social companionship. People tend to feel most supported when others are available and accessible [59], and when individuals ask for help and experience delays or no response, detrimental effects on experience and performance may emerge [60]. From the lens of media richness theory, instant messaging differs from the other forms of communication in this study by displaying high immediacy and feedback which allows it to accommodate high uncertainty [44]. In classroom research, synchronous chats have been shown to help foster a sense of community [61].

### 4.3. Correlates with Affective Organizational Commitment

Both average meaningfulness and email meaningfulness are related to affective organizational commitment. Affective organizational commitment is undergirded by a genuine emotional attachment and sense of connection to the organization [16]. Meaningfulness is also highly subjective, and possibly even derived from a sense of emotional connection, especially in social contexts [62]. To contrast, the effects of email meaningfulness may be spurious, as although it is possible that individuals who feel more connected to the organization experience emails as more meaningful, our correlation table shows that among all communication measures, meaningfulness of email relates significantly with both age and tenure.

### 4.4. Latent Modality Profiles

Our primary results are largely reinforced with the latent modality use profiles we extracted. Infrequent Communicators feel least connected, which is suggested in prior loneliness and isolation studies [33]. Burnout literature includes dimensions of both emotional exhaustion which is characterized as being drained by others, as well as depersonalization which refers to a sense of dehumanization or cynicism towards the people one works with or for [63]. Infrequent Communicators may already be disengaged or burned out, causing their sense of disconnection to lead to infrequent communication, as opposed to, or in concert with, the other way around. In contrast, in-person communicators had the highest meaningfulness ratings overall, which aligns with all of our other results and those of many prior researchers, which find in-person communication privileged above technological communication for a sense of connection [6,24].

Both remaining communication groups, Agile Communicators, and Modern Remote Communicators, had intermediate meaningfulness ratings statistically distinct from both the Infrequent Communicators and In-Person Communicators but not from each other, suggesting that while there may be predictable and differentiable patterns of communication choices, Agile Communicators use all modalities at a similar level, while the Modern Remote Communicators have no in-person time but the highest video call frequencies.

In many ways, this aligns with current thinking about hybrid and fully remote work schedules as the preferred "future" of work for many present-day employees [64]. The opportunity to connect with coworkers but also to spend more time at home seems to maximize an individual's overall well-being and sense of connection in a web of integrated work and life [64]. As work melds with home life, organizational researchers will likely want to include outcomes related to an individual's personal lives. Likely, the happiest and healthiest workforce of the future will be one for which they feel engaged, connected, and supported throughout the whole of their lives.

### 4.5. Limitations

As with all social science research, this study has limitations. First, the correlational nature of the study, while appropriate for our research question, cannot offer causational conclusions. In addition, this survey was all self-report and could suffer from common method bias [65,66]. If conducted in an applied setting, objective data regarding communication frequency and content could (and should) be analyzed in conjunction with self-report outcome scales to overcome this concern. In addition, we optimized our survey for reduced duration and cognitive load by removing modalities from non-work and meaningfulness questions which were recorded as being used 0 h per week by an individual. However, this resulted in the need for pairwise correlations and an inability to run certain statistics. A more robust survey could be designed to collect a response for every modality even if the workplace use is zero. This could open the door to predictive regression models which could identify the relative importance of frequency of use, non-work communication and meaningfulness by modality for each outcome. These variables then could also be included as predictors rather than outcomes in the latent profile analysis.

Finally, the study could have been more precise regarding its hypotheses and associated power and statistical corrections. Due to many possible effects and effect sizes, we chose a target sample size in the middle of a large range of possible outcomes, and then applied a Bonferroni correction across all pairwise comparisons. As a result, some large effect sizes were not discussed, and smaller effects of less-used modalities may not have been identified. It may be better to predict specific outcomes based on modalities and test those with appropriately sized random subsets of the data.

### 4.6. Future Directions

Future research should first and foremost address the limitations above. A field study which collected objective measures of frequency of use and non-work content in addition to the self-report measures would validate the use of self-report of those measures as well as the relationship of each with the subjective outcome measures.

In addition, experimental or qualitative research can and should be conducted to understand the apparent benefits of video calls and instant messaging. Research may well be conducted to continue enhancing the user experience of technological communication. Furthermore, the timing of media theory generation has lagged modern-day high-fidelity technologies. What is it exactly about face-to-face communication that generates connection, and can it be increasingly replicated in more technological settings? Even if technology never fully replicates an in-person experience, the efficacy of technologies like video calling suggest that user design research focused on outcomes of affective connection is likely to continue improving them. A balance of work-related and unrelated communications may be greatly beneficial across all mediums to enhance perceptions of job embeddedness, which is increasingly important through turbulent economic conditions that threaten job insecurity and turnover [67].

Regarding technology replacing in-person experiences, recent advancements in technology have demonstrated that interactions in virtual reality stimulate positive effects on feelings of loneliness [68]. As technology continues to develop, means of working and relevant forms of communication will develop in tandem. This is especially the case as virtual gathering spaces and their effects on communication meaningfulness and connectedness

continues to be studied [69]. Studying these modes of communication become more critical as the world becomes increasingly reliant on asynchronous means of communication, such as email and instant messaging, as opposed to synchronous means of communication, such as teleconferencing or phone calls [70].

## 5. Conclusions

Personal relationships benefit and bring meaning to individual experiences—when relationships at work are strong, individuals not only experience better mental and physical health, but also feel more connected to the organization they work for, put more energy into their work, and are more likely to stay with the organization longer. In a hybrid and virtual work world, it is imperative to understand how to use different forms of communication, not just to complete work, but also to keep employees meaningfully connected to each other and the organization.

In this study, we found that a sense of social companionship and meaningfulness of communication both have a lot of significant correlations among frequency of use, non-work, and meaningfulness ratings across modalities. More in-person communication often results in better affective outcomes, but hybrid and remote employees who stay in touch and connected to others still have positive outcomes that may be balanced out by other family-related outcomes from a flexible work arrangement.

In line with prior research, we confirm that modality use is complex and there may be no one-size fits all modality or theory to encompass what works: there is certainly more to understand in terms of the nuanced effects of different modalities for different people and in different circumstances, and how modality choice impacts outcomes or is impacted by the pursuit of certain outcomes. Nevertheless, it appears that hybrid and remote communication styles, provided they include some amount of in-person or video calling (i.e., rich technologies) generate moderately good affective organizational outcomes and may (although not included in the scope of this research study) generate overall more positive holistic life outcomes for employees. Moving forward, researchers and designers of human-computer interfaces should consider focusing on affective outcomes to inform their work. In the meantime, employers should understand that staying connected is vital to employee health and well-being, while the methods for doing so may vary. They should commit to listening to what is working and not working for their employees and be sure to provide numerous options to allow effective communication to arise.

**Supplementary Materials:** The following supporting information can be downloaded at: https://www.mdpi.com/article/10.3390/mti7110105/s1, survey used for this study is provided as part of the supplementary materials.

**Author Contributions:** Conceptualization, T.J.S. and D.R.S.; methodology, T.J.S. and D.R.S.; software, T.J.S. and D.R.S.; validation, T.J.S., D.R.S. and K.K.; formal analysis, T.J.S.; investigation, T.J.S.; resources, T.J.S. and D.R.S.; data curation, T.J.S.; writing—original draft preparation, T.J.S. and D.R.S.; writing—review and editing, T.J.S., D.R.S. and K.K.; visualization, T.J.S. and D.R.S.; supervision, D.R.S.; project administration, D.R.S. All authors have read and agreed to the published version of the manuscript.

**Funding:** This research received no external funding.

**Institutional Review Board Statement:** The study was conducted in accordance with the Declaration of Helsinki, and approved by the Institutional Review Board of San Francisco State University (protocol code 2021-169, approved on 2 June 2021).

**Informed Consent Statement:** Informed consent was obtained from all subjects involved in the study.

**Data Availability Statement:** The data presented in this study are openly available at https://zenodo.org/doi/10.5281/zenodo.10002698 (accessed on 29 October 2023).

**Conflicts of Interest:** The authors declare no conflict of interest.

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
