# Peer review of "Feeling Connected: Technology-Mediated Communication and the Relationship between Modality and Affective Outcomes"

_mti, doi:10.3390/mti7110105_

Round 1
Reviewer 1 Report
Comments and Suggestions for Authors
The authors of this manuscript researched the relationship between workplace communication and affective outcomes, and in particular the connectedness at work and affective organizational commitment. They consider the frequency of use across five communication modalities in relation to affective outcomes, and their relationships with communication meaningfulness. They completed an online survey to 516 participants, in which the participants self-reported weekly communication tendencies, experienced connectedness, and affective organizational commitment. Their findings include the following ones: generally, participants who reported more frequent communication at work reported higher levels of connectedness and higher affective organizational commitment; employees who found their communication more meaningful (irrespective of frequency) felt the most emotionally connected. They also discuss unique benefits of different communication modalities, as well as implications for hybrid and remote organizations.
Globally, the manuscript is very well written and organized.
There are some minor English issues that should be corrected; please refer to the attached commented PDF document where some of the needed corrections are highlighted.

Please refer to the comments above.
Author Response
Thank you for your time. Please see a point by point response in the attached document.

Reviewer 2 Report
Comments and Suggestions for Authors
The authors chose an interesting title for the manuscript. Abstract. There is no information on when the research was conducted. I recommend mentioning in this section the five communication modalities mentioned by the authors. It is recommended that the authors add information in the abstract in which enterprises the employed participants worked in (large, small, profit) and which country(s) the research concerns.
Introduction. This section lacks information about the novelty of the work and the gaps it fills.
In the methodology, section 2.3. Participants should definitely be expanded. There is no information on how participants were selected for the research or where they worked. I recommend writing a little more about Prolific Academic. In line 157, the authors write about requirements regarding the number of hours, but aren't the type of work (physical, mental), working conditions, and shift work taken into account here?
The methodology section lacks information on how the survey was constructed.
Please add the survey for viewing.
I recommend adding more references from 2022 and adding the latest items from 2023.
Comments on the Quality of English LanguageMinor editing of English language required
Author Response

(The authors gave the same response as above.)

Round 2
Reviewer 2 Report
Comments and Suggestions for Authors
The authors took into account all recommendations. The manuscript is readable, well-planned, and interesting.